# Genomics-Driven Precision Medicine in Pediatric Solid Tumors

**DOI:** 10.3390/cancers15051418

**Published:** 2023-02-23

**Authors:** Praewa Suthapot, Wararat Chiangjong, Parunya Chaiyawat, Pongsakorn Choochuen, Dumnoensun Pruksakorn, Surasak Sangkhathat, Suradej Hongeng, Usanarat Anurathapan, Somchai Chutipongtanate

**Affiliations:** 1Division of Hematology and Oncology, Department of Pediatrics, Faculty of Medicine Ramathibodi Hospital, Mahidol University, Bangkok 10400, Thailand; 2Department of Biomedical Science and Biomedical Engineering, Faculty of Medicine, Prince of Songkla University, Songkhla 90110, Thailand; 3Center of Multidisciplinary Technology for Advanced Medicine (CMUTEAM), Faculty of Medicine, Chiang Mai University, Chiang Mai 50200, Thailand; 4Pediatric Translational Research Unit, Department of Pediatrics, Faculty of Medicine Ramathibodi Hospital, Mahidol University, Bangkok 10400, Thailand; 5Musculoskeletal Science and Translational Research Center, Department of Orthopedics, Faculty of Medicine, Chiang Mai University, Chiang Mai 50200, Thailand; 6Translational Medicine Research Center, Faculty of Medicine, Prince of Songkla University, Songkhla 90110, Thailand; 7Department of Surgery, Faculty of Medicine, Prince of Songkla University, Songkhla 90110, Thailand; 8Division of Epidemiology, Department of Environmental and Public Health Sciences, University of Cincinnati College of Medicine, Cincinnati, OH 45267, USA

**Keywords:** precision medicine, pediatric solid tumor, actionable mutations

## Abstract

**Simple Summary:**

The detection of genomic aberrations in cancers has yielded a wealth of information to discover oncogenic drivers or pathogenic variants that are relevant for the development of precise treatment strategies. Recent studies have shown promising outcomes in adult cancer patients with well characterized cancer genetic biomarkers. However, the development of precise treatments for pediatric cancers is difficult due to the limited number of accessible samples and the fact that well-defined target genetic aberrations are limited. Here, we review the current landscape of pediatric precision oncology compared to adults and highlight the examples of single-arm and multiple-arm designs of pediatric precision treatments.

**Abstract:**

Over the past decades, several study programs have conducted genetic testing in cancer patients to identify potential genetic targets for the development of precision therapeutic strategies. These biomarker-driven trials have demonstrated improved clinical outcomes and progression-free survival rates in various types of cancers, especially for adult malignancies. However, similar progress in pediatric cancers has been slow due to their distinguished mutation profiles compared to adults and the low frequency of recurrent genomic alterations. Recently, increased efforts to develop precision medicine for childhood malignancies have led to the identification of genomic alterations and transcriptomic profiles of pediatric patients which presents promising opportunities to study rare and difficult-to-access neoplasms. This review summarizes the current state of known and potential genetic markers for pediatric solid tumors and provides perspectives on precise therapeutic strategies that warrant further investigations.

## 1. Introduction

Cancer occurrence before the age of 20 years is rare, but it is one of the leading causes of disease-related mortality in children and adolescents globally [1,2]. Approximately 300,000 children aged 0–19 years old worldwide are diagnosed with cancer each year [1], and 80% of these patients live in low- and middle-income countries (LMCs). Hematologic malignancies are more common among pediatric cancers, comprising about half of all cases. Solid malignancies are rarer and heterogenous as following an age-specific pattern. In early childhood, embryonal-type solid tumors are common, such as neuroblastoma, retinoblastoma, medulloblastoma, hepatoblastoma, and Wilms tumor [3]. The prognosis for childhood cancer has improved dramatically over the past four decades, particularly for hematologic malignancies [2]. Nonetheless, treatment outcomes for childhood solid malignancies remain unsatisfactory, especially in LMCs [4,5].

Genetic sequencing studies have led to the identification of somatic gene alterations as cancer hallmarks and germline predisposition and targeted the molecular abnormalities for the development of precise treatment [6,7,8]. Dramatic differences in the genetic repertoire between normal and cancer cells provide advantages of molecular targeted therapies over traditional strategies based on the target selectivity [9,10,11]. Several components in cellular signaling pathways, i.e., tyrosine receptor kinase (TRK), mitogen-activating protein kinase (MAPK) and phosphoinositide 3-kinases (PI3K)-mammalian target of rapamycin (mTOR), have been commonly identified as actionable mutations that would recommend appropriately targeted therapies [12,13]. These generic biomarker-driven precise treatments have been investigated in several pre-clinical and clinical trials since the early 2000s [14].

Progress in designing treatments targeting molecular alterations specific to pediatric cancers is considerably slow due to the rare and unique genetic alterations in children compared to adults [15]. A report from the European Union (E.U.) revealed that up to 26 anticancer drugs approved for adults might be also effective in pediatric malignancies; however, only four of these drugs have been approved for childhood cancers [16]. Nishiwaki S. and Ando Y. reported that only 3 out of 66 drugs with adult indications have been approved for pediatrics in the E.U., United States, and Japan [17]. Thus far, larotrectinib and entrectinib have been two of the most successful molecularly targeted therapies for children with solid tumors and have shown their promising responses in patients with NTRK-fusion [9]. In 2018, larotrectinib became the first drug to receive FDA approval to treat NTRK fusion-positive solid tumors in children and adults [18]. Similarly, entrectinib, a multi-kinase inhibitor, also received approval for the treatment of TRK fusion solid tumors in patients aged ≥ 12 years [19]. Combinatorial treatment of dabrafenib and trametinib has been recently approved by FDA (June 2022) for use in adult and pediatric patients > 6 years of age with unresectable or metastatic solid tumors with BRAF V600E mutation [New Drug Application (NDA): 202806 and 204114]. Note that abnormalities in *NRAS*, *ABL1*, *JAK2*, *KIT*, *ALK* and *BRAF* were among the group of common genetic variants found in adult and childhood cancers. In this review, we summarize the progress in the identification of actionable mutations in pediatric malignancies, FDA-approval status for pediatric and childhood treatment, and the recent update from clinical studies to explore the feasibility and utility of genomics-driven precision medicine.

## 2. Genetic Alterations on Cancer Hallmarks

### 2.1. Cancer Hallmarks and Common Targeted Signaling Pathways

Cancers are driven by changes in cellular DNA which further promote the transition of genetic landscape, especially in cell survival programs, leading to unstoppable cell growth with abnormal cellular characteristics [20]. In contrast to normal tissues, cancer cells can dysregulate their own signaling cascades autonomously, thus controlling their own cell fate [21]. Besides their proficiency in cancer hallmarks in evading growth suppressors, resisting cell death, reprogramming cellular mechanisms, and avoiding immune destruction, cancer cells can also acquire the capability to sustain proliferative signaling in several alternative ways [22,23]. Cancer cells may send signals to activate normal cells within the tumor parenchyma, which reciprocally communicate to supply cancer cells with various growth-promoting factors [24,25]. Furthermore, common downstream components in distinct signaling cascades also allowed cancer cells to control cell fate in a growth factor-independent manner by triggering the downstream molecules directly, negating the need for ligand-mediated receptor activation [23,26]. Hence, the vast majority of different cancers are coordinately modulated by canonical oncogenic drivers, including *KRAS*, *MYC*, *NOTCH*, and *TP53.* This factors highlights the need to fully elucidate their regulatory networks for further therapeutic development [27].

### 2.2. Tumor Cells Have Both Germline and Somatic Variants in Their Genome

Cancer gene mutations can be either inherited or acquired. Hereditary or germline mutations refer to the genomic changes that occur in germ cells and can be detected in all cells of the offspring and are passed inter-generationally [28,29]. Genetic predisposition has been described by certain characteristics, including [30];

Familial history of the same or related cancers;Occurrence of bilateral or multifocal cancers;Earlier age at disease onset;Physical suggestive of a predisposition syndrome;Appearance of specific tumor types corresponding to the genetic predisposition.

Several studies have described germline mutations in cancer including *BRCA1/2*, *TP53*, *ATM*, *CHEK2*, *MSH2* and *PALB2* [31,32,33]. Cancer cells harboring these germline predispositions are prone to increase cancer susceptibility, developing cancers at younger ages than usual. Using the 565 cancer-predisposing gene (CPG) panel for germline mutation analysis in children and adolescents with pan-cancer (*n* = 1120), Zhang et al. [31] reported that 95 pathogenic variants were detected in 21 of the 60 autosomal dominant CPGs in 94/1120 patients. Interestingly, the prevalence of germline mutation was greatest among patients with non-CNS solid tumors (16.7%), followed by brain tumors (8.6%) and leukemia (4.4%) [31]. Genetic predisposition syndromes associated with rare cancers of pediatric solid malignancies are provided in Table 1 [34,35,36]. Cancer predisposition syndrome such as Li–Fraumeni syndrome (LFS) with *TP53* mutation generally promotes the onset of various benign and malignant neoplasms, such as neuroblastoma (NB), osteosarcoma (OS), soft tissue sarcomas (STS), and brain tumors [37]. Mutations in *NF1* are associated with neurofibromatosis (NF), low- and high-grade gliomas (L/HGGs), and malignant peripheral nerve sheath tumors. Mutations in *SUFU* or *PTCH1* in Nevoid basal cell carcinoma are relevant to the development of the sonic hedgehog (SHH) subgroup-medulloblastoma (MB) [38].

Somatic mutations are de novo genetic alterations that spontaneously develop in an individual cell over time and play a vital role in cancer development and progression [51]. Studies have shown that the number of genetic abnormalities identified in each cancer patient may increase over time, leading to tumor survival against the selective pressure of drug actions, thereby acquiring resistance and causing disease progression [13,52]. Commonly identified somatic mutations include those involved in RTK signaling (*PDFGRA*, *ERBB2* and *EGFR*), MAPK signaling (*NF1*, *KRAS*, and *MAP2K1*), PI3K-mTOR signaling (*PIK3CA*, *MTORC1/2* and *PTEN*), cell cycle (*CDKN2A/B*, *RB1* and *ATM*), DNA maintenance (*TP53*), transcriptional regulators (*MYC* and *MYCN*), and epigenetic modifiers (*SMARCB1* and *ATRX*) [12,53]. Cancers usually involve a different spectrum of mutation which are strongly associated with pathogenesis and disease prognosis. A pan-cancer analysis reported by Grobner et al. [33] showed that 93% of adult cancer patients harbor at least one significantly mutated gene, while only 47% presented such mutations in pediatric tumors. However, approximately 30% of recurrent hot-spot mutations in pediatrics overlapped with adult cancers, highlighting some potential druggable targets based on finding from adult cancers. Hence, advances in identifying and understanding oncogenic drivers and actionable mutations would further improve the current therapeutic strategies for the development of precision medicine in cancers.

### 2.3. Germline and Somatic Variants Classified as Druggable

In the context of defining mutational actionability, the relevant effects of genomic aberration participating in cancer phenotypes are considered. DNA aberrations include missense, nonsense, frameshift mutations, and chromosome rearrangements, with some changes affecting only a single DNA base that may or may not alter the protein’s property and some point mutations completely abrogating protein expression. A wide variety of gene alterations have been detected such as activating point mutation in *BRAF*, *ALK*, *EGFR* and *FGFR1* genes, high copy number gains in *PDGFRA* and *ERBB2*, loss-of-function mutation affecting *PTEN*, *PTPN11*, *PIK3R1*, and *MTORC1*, *CDKN2A/2B* deletions, or in-frame expression of large indels (*NOTCH1* and *FOXA1*) [12]. Other changes involving larger stretches of DNA may include rearrangements, deletions, or duplications of long stretches of DNA [54]. For example, exon skipping on MET exon 14 proto-oncogenes resulting from intronic mutation increases the protein lifespan and promotes MET activation in lung carcinogenesis [55].

The significance of genetic variants may vary depending upon their potential effects on cellular functions. An “actionable” mutation is defined as a genetic aberration that is potentially responsive to targeted therapy, while a “driver” mutation refers to variants that confer a growth advantage to cancer cells but may not be targetable with a specific treatment yet. Passenger mutation is used to designate cancer-neutral variations and is unlikely to be under selective pressure during the evolution of the cancerous cells [56,57]. The “passenger” mutation has the lowest tendency to impact protein function, most of which are synonymous substitutions; however, these mutations occur more frequently than driver or actionable mutations. Unraveling the passenger mutational paradigm has otherwise revealed the existence of pre-existing latent driver mutations in which certain combinations of the passenger mutations could indeed be functional drivers. One example is the non-hotspot, passenger mutation of the *Akt1* gene at position L52R, C77F, and Q79K, which promotes its membrane localization similarly to the E17K driver. In contrast, the co-existence of D32Y, K39N, and P42T passenger mutations can lead to Akt conformational inactivation, suggesting that treatment decisions based only on genetics may overlook crucial actionable components [56,58]. In addition, silent mutations occurring near the donor splice junction could contrarily affect exon splicing. For example, T125T mutation in *TP53* is a recurrent mutation that is generally considered a non-functional passenger event; however, its existence at the −1 donor site of exon 4 raises the possibility that this mutation affects splicing. Further integration with RNA-seq data demonstrated that T125T mutation resulted in the retention of intron 4 and introduced a premature stop codon such as nonsense-mediated decay [59]. Thus, aberrant splicing caused by silent mutations should be carefully evaluated during interpretation of the sequencing results.

The accumulated data of genetic composition data from the tumors of patients has become a growing compendium of molecular biomarkers for precise treatment with FDA-approved drugs. Figure 1 summarizes the actionable mutations currently approved by FDA consortium for targeted therapy in adult cancers and pediatric solid tumors. Common actionable genetic aberrations associated with the National Comprehensive Cancer Network (NCCN) guidelines or FDA-approved targeted therapies are extensively summarized in Table 2. The data were predominantly gathered from the OncoKB database and the representative cancer types, and levels of evidence were included [60].

## 3. Pediatric Cancer Genome

### 3.1. Pediatric vs. Adult Cancer Development

Pediatric cancers reflect a heterogeneous group of disorders distinct from adult cancers in terms of cellular origins, genetic complexity, and specific driver alterations [62,63]. Pediatric malignancies typically occur in developing mesoderm rather than adult epithelia (ectoderm) and are often induced by inherited or sporadic errors during development [33]. Studies have quantified the mutation burden in many pediatric cancers, identifying approximately 5 to 10 protein-coding variants identified across multiple tumor types except in osteosarcoma, which showed an average of 25 protein-affecting mutations. In contrast, the average number of mutations in adult cancers ranges between 33 to 66 in pancreatic, colon, breast, and brain cancers while mutagen-caused adult tumors (such as melanoma and lung cancers) can include up to 200 protein-coding variants [64,65,66]. At diagnosis, patients with pediatric cancers tend to have less complexity on mutational spectra than those in adult cancers; however, with treatment-refractory tumors and recurrence—the mutation rates in pediatric tumors have increased to be comparable to adult tumors [67,68]. Moreover, the rare occurrence of pediatric cancers and the low frequency of recurrent genomic alterations have a great impact on the investigations and the availability of targeted agents. Thus, there is an urgent need to accelerate the pace of genomic data acquisition and clinical trials in children to design more effective strategies for pediatric precision oncology.

### 3.2. Somatic and Germline Mutations Identified in Pediatric Cancer Cohorts

Single nucleotide variations (SNVs) and small indels are the usual mutations identified in adult cancers. In contrast, childhood cancers show a relatively high prevalence of copy number aberrations (CNAs) and specific structural variations (SVs). Note that insertion and deletion lead to adding and removing at least one nucleotide to the gene, respectively, which can affect protein functions and contribute to carcinogenesis. Current data suggest that approximately 10% of pediatric cancers are caused by genetic predisposition [32]. Zhang et al. [31] revealed that 95 out of 1120 (8.5%) patients younger than 20 years of age harbor germline mutations in cancer-predisposing genes. Diets et al. [69] performed trio-based whole-exome sequencing on the germline DNA of 40 selected children with cancer and their parents. Of these, germline pathogenic mutations were identified in 20% (8/40) of children with cancer [69]. Similarly, Grobner et al. [33] reported that most germline variants were related to DNA repair genes from mismatch (MSH2, MSH6, PMS2) and double-stranded break (TP53, BRCA2, CHEK2) repair.

Using combined somatic and germline sequencing for children with solid tumors, Parsons et al. [32] identified actionable mutations in up to 40% (47/121) of pediatric solid tumor tissues. Likewise, Wong et al. [12] performed the combination of tumor and germline sequencing (WGS) and RNA sequencing (RNA-seq) to identify 968 reportable molecular aberrations (39.9% in both WGS and RNA-seq; 35.1% in WGS only and 25.0% in RNA-seq only) in 247 high-risk pediatric cancer patients with 252 tumor tissues. Interestingly, 93.7% of these patients had at least one germline or somatic aberration, 71.4% had therapeutic targets, and 5.2% had a change in diagnosis [12].

These cohort studies emphasized that comprehensive molecular profiling could resolve molecular aberration in high-risk pediatric cancer and provide clinical benefits in a significant number of patients. In the era of next-generation sequencing, publicly genomic data access is considered one of the keys to accelerate research. The St. Jude Cloud is one of the most promising data-sharing ecosystems, with genomic data from >10,000 pediatric patients with cancer and long-term survivors. When exploring the mutational profile of pediatric solid tumors, the resource has revealed common genetic alterations among the different cancer types, as shown in Table 3. This integrative view of genomic data could be further used to expedite studies of pediatric cancer-associated risk factors and initiate novel therapeutic investigations for improving treatment outcomes.

### 3.3. Predictive and Common Genetic Variant Abnormalities Identified in Pediatric Tumors

The reports of actionable mutations identified in various studies have ranged from 27% to 100%, depending on the study design [6]. Several methods have been adopted for comprehensive molecular analysis to discover the actionable mutations that result in the targeting of cancer-associated elements. Table 4 contains a comprehensive, up-to-date summary of genomic aberrations found in pediatric solid tumors, together with potential targeted treatments, based on several public databases [60,70,71,72,73]. We systemically reviewed genomic alterations with high prevalence in pediatric cancers using comprehensive WES and RNA-seq data via the St. Jude Cloud (www.stjude.cloud; accessed on 26 September 2022) [70]. Importantly, the genomic point mutations and gene fusions reported by this public domain are unique and different from those variants identified in the OncoKB database (the mutational collection of adult cancers) [60]. In addition, the potential druggable targets of these significant genomic alterations required further testing in pediatric solid tumor patients. A significant number of studies [60,69,70,71,72,73,74,75,76,77,78,79,80,81,82,83,84,85] were reported by the Clinical Interpretation of Variants in Cancer (CIViC) database (https://civicdb.org; accessed on 18 September 2022) [71] which matched genomic alteration and molecularly targeted therapies tested in pediatric patients. These treatment designs were translated from the clinical care of adults across different tumor types but harboring the same genetic dysregulation, which gave satisfactory clinical outcomes. For pediatric solid tumors with no clinical evident support or undruggable genomic alterations, we listed the potential targeted therapies based on the knowledge from adult cancers as suggested by cBioPortal (www.cbioportal.org; accessed on 30 April 2022) [72,73] and OncoKB (https://www.oncokb.org; accessed on 17 April 2022) [60] that should be considered for further investigation and optimization for pediatric treatments. As of now, fewer number of patients could hinder the availability of molecular characterization and statistically meaningful preclinical/clinical outcomes. However, this challenge can be overcome by the initiation of multi-institutional cooperation and international data sharing, which would enable clinicians to effectively explore optimized therapeutic interventions toward pediatric precision oncology.

## 4. Current Progress in Clinical Trials for Pediatric Precision Oncology

Genomic precision medicine has demonstrated preferential outcomes among ongoing genomic-driven clinical trials in adult cancers. Yet, clinical investigations based on pediatric tumor genetics are still lacking. Based on the patient genetic profile screening, scattered reports on molecularly defined pediatric patients are showing prominent responses to some targeted therapies. For example, targeting *ALK* has shown success in treatments of ALK(+) non-small cell lung cancers and also in childhood anaplastic large cell lymphoma (ALCL) and inflammatory myofibroblastic tumor using the ALK inhibitor crizotinib [92]. While *ALK* mutation is the most common somatic mutation in neuroblastoma, crizotinib was compromised due to the interference by common *ALK* mutation F1174 [93]. Since then, ceritinib, alectinib, brigatinib, and lorlatinib have been approved against advanced *ALK*+ NSCLC [94,95,96,97]. Intriguingly, the third-generation TKI that targets both ALK and ROS1, lorlatinib, has recently shown promise in patients with *ALK* mutated neuroblastoma, but most of the studies are still at phase I clinical trial. [98]. Nonetheless, repotrectinib, a next-generation ROS1/TRK inhibitor with >90-fold potency against ROS1 than crizotinib in NSCLC patients is also being tested for dose escalation in phase II clinical trial with patients aged ≥ 12 years [99]. Another promising example is the targeted therapy against Ras-Raf-MEK-ERK signaling cascade which include somatic *BRAF* alterations (*BRAF* V600E and *BRAF* fusions). The prototype for targeting *BRAF* V600E/K is cutaneous melanoma, where 40–60% of patients with these mutations are eligible for the FDA-approved BRAF-inhibitor, vemurafenib [100]. Low-grade-gliomas have been identified to contain multiple alterations in Ras-Raf-MEK-ERK pathway, and a single treatment of vemurafenib in malignant glioma resulted in tumor regression [85,101]. Recently, Jain et al. [102] reported that a combination of BRAF-inhibitor dabrafenib and MEK-inhibitor trametinib enhanced treatment efficacies in pediatric low-grade-glioma carrying *KIAA1549-BRAF* fusion. Additionally, several studies have utilized the combination of molecularly targeted agents and traditional chemotherapy or radiation to reduce the severe side effects caused by an intensive dose of chemo/radiotherapy while minimizing acquired drug resistance due to selective pressure (Table 5).

The following large-scale pediatric and young-adult precision oncology programs have been launched with multiple-arm trials for patients with matched molecular profiles: TAPUR (ClinicalTrials.gov identifier NCT02693535), NCI-COG Pediatric MATCH (NCT03155620), the Tumor-Agnostic Precision Immuno-Oncology and Somatic Targeting Rational for You (TAPISTRY) (NCT04589845). These global, multicenter, open-label, multi-cohort studies are now at phase II, and the treatment assignment has relied on the basis of relevant onco-genotypes as identified by a Clinical Laboratory Improvement Amendments (CLIA)-certified or a validated next-generation sequencing (NGS) assay. While the eligible criteria of TAPUR are open for patients aged 12 years old or older, most of the patients enrolled are reported to have adult cancer phenotypes [103,104,105]. In contrast, the NCI-COG Pediatric MATCH aims to evaluate the molecular-targeted therapies with selected biomarkers of childhood and young adult patients with a reported detection rate of actionable alterations of 31.5% from the first 1000 tumors screened. Assignments to treatment arms were made for 28% of patients screened and 13% of patients enrolled in the treatment trial [106]. In the TAPISTRY study, nine targeted treatments are being examined, and eleven non-randomized treatment arms are available for participants of all ages with locally advanced/metastatic solid tumors. The purpose of this study is to evaluate the safety and efficacy of different targeted therapies and immunotherapies in patients as single agents, but the results of the study are still to be released. Overall, the advancements in high-throughput sequencing technology have closed the gap between the current treatment paradigm and precision medicine, markedly improving rates of response, progression-free survival (PFS), and overall survival (OS) compared to traditional randomized trials. Moreover, the multicenter, open-label, multi-arm treatment designs can further benefit treatment strategies by yielding efficacy and toxicity data in a timely manner with cost-effectiveness. Therefore, in the future, international coordination will be crucial to generate a database to inform rational trial design and to evaluate the combination of treatments/interventions that ensure more favorable outcomes.

The current applications of precision study designs for pediatric cancers (summarized from clinicaltrials.gov; accessed on 17 August 2022) are shown as single-arm and multiple-arm designs in Table 5 and Table 6, respectively.

## 5. Challenges and Perspectives

Large-scale cancer sequencing studies such as the 1000 Genomes Project [107], The Cancer Genome Atlas (TCGA) [108], and the International Cancer Genome Consortium (ICGC) [109] provide an extensive landscape of tumor genomic profiles which substantially facilitate the predication of recurrent hot-spot mutations on the selected type of cancers. Other large databases aim to collect the profile of childhood cancers include St. Jude/Washington University Pediatric Cancer Genome Project (PCGP) [110] and NCI’s Therapeutically Applicable Research to Generate Effective Treatments (TARGET) [53] which are accessible via the St. Jude Cloud (https://www.stjude.cloud, accessed on 26 September 2022) public data repository. These large-scale studies have confirmed that the spectra of genomic alterations and their relevant mechanisms differ in childhood tumors from those predominantly occurring in adult cancer—at least by half. Thus, the actionability of pediatric-driven mutations needs to be carefully interpreted before translating into a targeted treatment option.

Several challenges need to be addressed when researchers launch the study/trial for pediatric cancer treatment. Many pediatric cancers are rare, and finding the right patient population for the drugs is challenging. In fact, a small patient population and a prolonged trial duration are not uncommon issues in the settings of rare diseases and low-incidence pediatric cancers [111,112,113,114]. Optimal statistical designs for less stringent comparisons, for example, by relaxing type I error (higher than 5%) or power (lower than 80%) can still provide meaningful results from small but faster trials [111,112,113,114]. Implementing multi-arm multi-stage trial design would allow patients with poor prognosis to be stratified into multiple phase II arms; receiving the window-of-opportunity/experimental therapies and restaging by serial biopsies and molecular characterizations to inform ongoing treatment choices [113,114]. These approaches remain useful to increase the overall feasibility for rare disease trials, i.e., keeping the sample size as small as possible while maintaining the power and ability to address the trial objectives.

Only 45% of pediatric cancer driver genes are shared with adult cancers, suggesting that novel therapeutic agents are required for pediatric cancer. Additionally, pediatric cancers are often driven by structural variants that can be challenging to identify and target. Nonetheless, children with cancers have accumulated fewer genetic mutations, thus making genomic targeting simpler than adults [113]. In a broad view, cancer intrinsic targets (e.g., mutated oncogene, tumor suppressor, epigenetics, synthetic lethal, and DNA damage) play crucial roles in cancer pathogenesis and thus could serve as the key stones for drug development against childhood cancers [115]. Another approach in drug development strategy is a mechanisms-of-action (MoA)-driven approach which successfully exemplified the efficiency of nivolumab and larotrectinib as targeted anticancer drugs against programmed cell death protein-1 (PD1) and TRK receptors, respectively [116]. Nonetheless, lessons learned from adult cancers have warned us that many pediatric cancers would have failed to express mutated kinase targets, and resistance to targeted therapies would rapidly occur. Recently, newly emerging cancer targets have been discovered upon multidimensional complexity of the dynamic oncogenic states, for example, tumor archetypes, master regulators, cancer-associated protein–protein interactions, and metabolic vulnerabilities [115,117,118,119,120]. The development of drugs against the emerging classes of cancer targets may deliver adjunct/complementary agents for combination with targeted therapeutic regimens [115]. The emergence of gene editing technologies such as transcription activator-like effector nucleases (TALENS) and clustered regularly interspaced palindromic repeats (CRISPR) paired with the CRISPR-associated endonuclease 9 (CRISPR-CAS9) offer the powerful customizable therapeutic options to precisely edit the targeted genes [121,122,123], thus providing hope to all pediatric cancers to be benefited from genomic-driven precision medicine approach.

Comprehensive molecular profiling of the genetic variants/mutations, gene expression at both transcripts and protein levels, and perhaps information on post-translational modifications and metabolites are coordinately utilized to improve the accuracy of molecularly targeted agents. Challenges in this grand scheme, besides big data sharing and multi-omics integration, are interpreting complex high-dimensional data in the biological sense, prioritizing findings into actionable targets/pathways, and achieving the candidate compounds/drugs for precise treatment. Aberrant expression of messenger RNA associated with genomic changes could contribute to the biology of tumor progression. In most cases, RNA-seq analysis can increase the coverage number of variant curations, especially the comprehensive gene fusion discovery and tumor expression subgroup analysis, when compared to WGS alone [124]. A novel molecularly guided approach, so-called transcriptomic connectivity analysis, utilizes the power of RNA-seq to detect aberrant gene expression and employs transcriptomic reversal of cancer cells/tissues for repurposing FDA-approved drugs [125,126,127]. This molecularly guided therapeutic approach could be an asset for prioritizing the approved drugs for off-label use in childhood cancer trials.

Despite the promising demonstration of ongoing genomic-driven clinical trials of targeted anticancer small molecules, cancer immunotherapies have become significant advances for pediatric solid tumors [128,129]. Ganglioside GD2 is a sialic acid-containing glycosphingolipid that highly expressed on the surface of multiple pediatric solid tumors, i.e., neuroblastoma, osteosarcoma, Ewing sarcoma, rhabdomyosarcoma, and brain tumors including diffuse intrinsic pontine glioma (DIPG) and medulloblastoma [128,129]. Thus, GD2 is recognized as one of the most promising targets for pediatric cancer immunotherapy. Dinutuximab, anti-GD2 monoclonal antibody, has been approved as the first-line therapy for high-risk pediatric neuroblastoma [128,129,130], while GD2-specific chimeric antigen receptor (CAR) T cell therapy is under investigation in the early phase trials for children with neuroblastoma, osteosarcoma, and brain tumors (ClinicalTrials.gov identifier NCT03721068, NCT04539366, NCT04099797, NCT04196413). Besides GD2, newly emerging targets for pediatric cancer immunotherapy, including PD1/PD-L1 (NCT04544995, NCT04796012), B7-H3 (CD276; NCT04864821, NCT04743661), HER2 (NCT00902044, NCT04616560) and CD47 (NCT04525014, NCT04751383), have been actively investigated for pediatric sarcomas and brain tumors.

Last but not least, it should be noted that new therapeutics often lack dosage guidelines for children [12]. Acknowledging children have different drug responses and tolerance profiles compared to adults, it is crucial to define the optimal dosages of new drugs/biologics (and the off-label use of FDA-approved medications) to achieve preferred therapeutic outcomes. Recent innovations in study designs (i.e., phase I dose-finding design for pediatric population, the potential inclusion of children in adult trials, cooperative group trials) [131,132,133,134], together with the regulatory initiatives in the United States (US) and the E.U. which encourage the development of novel anticancer therapies in children [134,135], provide guidance to address this challenge while accelerating the pace of genomic-driven precision medicine in pediatric oncology.

## 6. Conclusions

Essential questions that need to be addressed in applications of precision therapeutic program include the applicability of the genetic testing, the significance of the mutation variant, and the existence of an approved targeted therapy. Although targeted agents are approved for a set of tumors harboring specific mutations, future development of clinical guidelines may recommend these agents to be used off-label in different tumor types with the same mutations. Identifying the mutational signatures of pediatric solid tumors will open opportunities for new targeted therapeutic strategies since their malignant origin manifests differently from the adults. Similar genomic-driven precision medicine approaches have been launched by several institutes, while the long-term effects of many of those novel agents are just beginning to be evaluated. These treatments could improve survival and reduce toxicity in pediatric patients and maximize therapeutic advantages when incorporated into standard care.

## Figures and Tables

**Figure 1 cancers-15-01418-f001:**
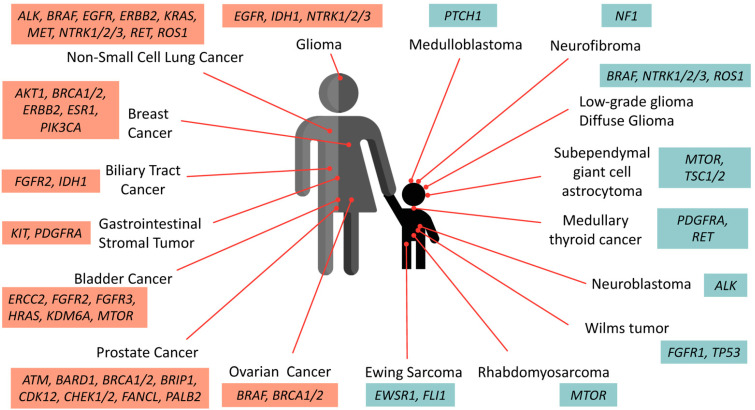
Oncogenic drivers identified in adult and pediatric solid tumors. These selective biomarkers are predicted to be responsive to various levels of FDA-approved drugs (detailed in Table 1). Note that targeted therapies against *PTCH1* and *ALK* in medulloblastoma and neuroblastoma are currently undergoing clinical assessment and awaiting further approval.

**Table 1 cancers-15-01418-t001:** Mutated genes and dysregulated signaling pathways in selected cancer predisposition syndromes.

Cancer Predisposition Syndrome	Common Solid Tumors	Mutated Genes (Inheritance)	Dysregulated Pathways	Reference
Beckwith–Wiedemann syndrome	Wilms tumor, hepatoblastoma, neuroblastoma, rhabdomyosarcoma	*CDKN1C* (AD)	Cell cycle	[39,40]
Constitutional mismatch repair deficiency	Brain tumor, neuroblastoma, Wilms tumor, osteosarcoma, rhabdomyosarcoma	*MLH1*, *MSH2*, *MSH6*, *PMS2* (AR)	DNA mismatch repair	[36,41]
Hereditary retinoblastoma	Retinoblastoma, melanoma, osteosarcoma, pineoblastoma	*RB1* (AD)	Cell cycle	[39,42]
Li-Fraumeni syndrome	Brain tumor, sarcoma, neuroblastoma, rhabdomyosarcoma, retinoblastoma	*TP53* (AD)	Cell cycle, apoptosis	[39,43,44]
Neurofibromatosis	Glioma, astrocytoma, ependymoma, malignant peripheral nerve sheath tumors, neuroblastoma, rhabdomyosarcoma	*NF1*, *NF2* (AD)	RAS/MAPK	[39,45]
Rhabdoid tumor predisposition syndrome	Atypical teratoid/rhabdoid tumor, malignant rhabdoid tumor	*SMARCB1*, *SMARCA4* (AD)	Wnt/*β*-catenin, Sonic hedgehog	[39,46]
Multiple endocrine neoplasia	Ependymoma, Medullary thyroid cancer	*MEN1*, *RET* (AD)	Transcriptional activity	[39,47]
Nevoid basal cell carcinoma	Medulloblastoma, rhabdomyosarcoma	*PTCH1*, *PTCH2*, *SUFU* (AD)	Sonic hedgehog	[39,46]
Familial adenomatous polyposis	Medulloblastoma, hepatoblastoma	*APC* (AD)	Wnt/*β*-catenin	[39,48]
Tuberous sclerosis	Subependymal giant cell astrocytoma, rhabdomyosarcoma	*TSC1*, *TSC2* (AD)	mTOR	[39,49]
Bloom syndrome	Osteosarcoma, Wilms tumor	*BLM* (AR)	DNA double-strand repair	[34,35]
Rubinstein–Taybi syndrome	Medulloblastoma, neuroblastoma, rhabdomyosarcoma	*CREBBP* (AD)	Transcriptional regulation	[34,35]
Noonan syndrome	Rhabdomyosarcoma, neuroblastoma, glioma, hepatoblastoma	*PTPN11*, *SOS1*, *RAF1*, *KRAS*, *MAP2K1* (AD)	RAS/MAPK	[50]

Abbreviations: AD, autosomal dominant; AR, autosomal recessive.

**Table 2 cancers-15-01418-t002:** Targeted therapies recommended for the selected genetic alterations according to FDA-approved or NCCN guidelines [60].

Gene	Alterations	Targeted Therapies	Cancer Types	FDA-Approved Level ^a^
*AKT1*	E17K	AZD5363	Breast Cancer, Ovarian Cancer; Endometrial Cancer	Lv.3
*ALK*	Fusions	Alectinib; Brigatinib; Ceritinib; Crizotinib	Non-Small Cell Lung Cancer	Lv.1
		Brigatinib; Ceritinib; Crizotinib	Inflammatory Myofibroblastic Tumor	Lv.2
	Oncogenic Mutations	Lorlatinib	Non-Small Cell Lung Cancer; Neuroblastoma ^c^	Lv.1
		Crizotinib	Non-Small Cell Lung Cancer; Neuroblastoma ^c^	Lv.R2
*ARAF*	Oncogenic Mutations	Sorafenib	Non-Small Cell Lung Cancer	Lv.3
*ARID1A*	Truncating Mutations	PLX2853; Tazemetostat	All Solid Tumors	Lv.4
*ATM*	Oncogenic Mutations	Olaparib	Prostate Cancer	Lv.1
*BRAF*	V600E	Dabrafenib + Trametinib	Melanoma; Non-Small Cell Lung Cancer; Low grade glioma ^b^; High grade glioma ^b^	Lv.1
		Encorafenib + Cetuximab	Colorectal Cancer	
	Fusions or V600E	Selumetinib	Pilocytic Astrocytoma	Lv.2
	V600E	Dabrafenib + Trametinib, Vemurafenib + Cobimetinib	Diffuse Glioma; Encapsulated Glioma; Ganglioglioma	
	Fusions	Trametinib; Cobimetinib	Ovarian Cancer	Lv.3
	V600E	Dabrafenib + Trametinib	Biliary Tract Cancer	
	G464, G469A, G469R, G469V, K601, L597	PLX8394	All Solid Tumors	Lv.4
*BRCA1/2*	Oncogenic Mutations	Niraparib; Olaparib; Olaparib + Bevacizumab; Rucaparib	Ovarian Cancer; Peritoneal Serous Carcinoma	Lv.1
		Olaparib; Rucaparib	Prostate Cancer	
		Olaparib; Talazoparib	Breast Cancer	Lv.3
*BRIP1*	Oncogenic Mutations	Olaparib	Prostate Cancer	Lv.1
*CDK4*	Amplification	Palbociclib; Abemaciclib	Dedifferentiated Liposarcoma; Well-Differentiated Liposarcoma	Lv.4
*CDK12*	Oncogenic Mutations	Olaparib	Prostate Cancer	Lv.1
*CDKN2A*	Oncogenic Mutations	Palbociclib; Ribociclib; Abemaciclib	All Solid Tumors	Lv.4
*CHEK1/2*	Oncogenic Mutations	Olaparib	Prostate Cancer	Lv.1
*EGFR*	Exon 19 deletion, L858R	Afatinib; Dacomitinib; Erlotinib; Erlotinib + Ramucirumab; Gefitinib; Osimertinib	Non-Small Cell Lung Cancer	Lv.1
	Exon 20 insertion	Amivantamab; Mobocertinib		
	G719, L861Q, S768I	Afatinib		
	T790M	Osimertinib		
	A763_Y764insFQEA	Erlotinib		Lv.2
	E709_T710delinsD	Afatinib		Lv.3
	Exon 19 insertion	Erlotinib; Gefitinib		
	Exon 20 insertion	Poziotinib		
	Kinase Domain Duplication	Afatinib		
	A763_Y764insFQEA or Exon 19 insertion or L718V, L747P	Afatinib		Lv.4
	D761Y	Osimertinib		
	Kinase Domain Duplication	Erlotinib; Gefitinib		
	Amplification or A289V, R108K, T263P	Lapatinib	Glioma	
	Exon 20 insertion, T790M	Erlotinib; Gefitinib; Afatinib	Non-Small Cell Lung Cancer	Lv.R1
	C797S, D761Y, G724S, L718V	Osimertinib; Gefitinib		Lv.R2
*ERBB2*	Amplification	Ado-Trastuzumab; Emtansine; Lapatinib + Capecitabine; Lapatinib + Letrozole, Margetuximab + Chemotherapy; Neratinib; Neratinib + Capecitabine; Trastuzumab + Pertuzumab + Chemotherapy; Trastuzumab + Tucatinib + Capecitabine; Trastuzumab Deruxtecan; Trastuzumab, Trastuzumab + Chemotherapy	Breast Cancer	Lv.1
		Pembrolizumab + Trastuzumab + Chemotherapy; Trastuzumab + Chemotherapy; Trastuzumab Deruxtecan	Esophagogastric Cancer	Lv.1
		Trastuzumab + Lapatinib; Trastuzumab + Pertuzumab; Trastuzumab Deruxtecan	Colorectal Cancer	Lv.2
	Oncogenic Mutations	Ado-Trastuzumab; Emtansine; Trastuzumab Deruxtecan	Non-Small Cell Lung Cancer	Lv.2
		Neratinib	Breast Cancer; Non-Small Cell Lung Cancer	Lv.3
*ESR1*	Oncogenic Mutations	AZD9496; Fulvestrant	Breast Cancer	Lv.3
*FANCL*	Oncogenic Mutations	Olaparib	Prostate Cancer	Lv.1
*FGFR1*	Amplification	Debio1347; Infigratinib; Erdafitinib	Lung Squamous Cell Carcinoma	Lv.3
	Oncogenic Mutations	Debio1347; Infigratinib; Erdafitinib; AZD4547	All Solid Tumors	Lv.4
*FGFR2*	Fusions	Erdafitinib	Bladder Cancer	Lv.1
		Infigratinib; Pemigatinib	Cholangiocarcinoma	
	Oncogenic Mutations	Debio1347; Infigratinib; Erdafitinib; AZD4547	All Solid Tumors	Lv.4
*FGFR3*	Fusions or G370C, R248C, S249C, Y373C	Erdafitinib	Bladder Cancer	Lv.1
	G380R, K650, S371C	Erdafitinib		Lv.3
	Oncogenic Mutations	Debio1347; Infigratinib; Erdafitinib; AZD4547	All Solid Tumors	Lv.4
*FLI1*	EWSR1-FLI1 Fusion	TK216	Ewing Sarcoma	Lv.4
*HRAS*	Oncogenic Mutations	Tipifarnib	Bladder Urothelial Carcinoma; Head and Neck Squamous Cell Carcinoma	Lv.3
*IDH1*	R132	Ivosidenib	Cholangiocarcinoma	Lv.1
	Oncogenic Mutations		Chondrosarcoma	Lv.2
	R132		Glioma	Lv.3
*KDM6A*	Oncogenic Mutations	Tazemetostat	Bladder Cancer	Lv.4
*KIT*	A502_Y503dup, K509I, N505I, S476I, S501_A502dup, A829P and 5 other alterations, D572A and 65 other alterations, K642E, T670I, V654A	Imatinib; Regorafenib; Ripretinib; Sunitinib	Gastrointestinal Stromal Tumor	Lv.1
	A829P and 5 other alterations	Sorafenib	Gastrointestinal Stromal Tumor	Lv.2
*KRAS*	G12C	Sotorasib	Non-Small Cell Lung Cancer	Lv.1
		Adagrasib	Non-Small Cell Lung Cancer	Lv.3
		Adagrasib; Adagrasib + Cetuximab	Colorectal Cancer	
	Oncogenic Mutations	Cobimetinib; Trametinib; Binimetinib	All Solid Tumors	Lv.4
*MAP2K1*	Oncogenic Mutations	Cobimetinib; Trametinib	Melanoma; Non-Small Cell Lung Cancer; Low grade glioma ^c^	Lv.3
*MDM2*	Amplification	Milademetan	Dedifferentiated Liposarcoma; Well-Differentiated Liposarcoma	Lv.4
*MET*	D1010, Exon 14 deletion, Exon 14 splice mutation	Capmatinib; Tepotinib	Non-Small Cell Lung Cancer	Lv.1
	Amplification or D1010, Exon 14 deletion, Exon 14 splice mutation	Crizotinib		Lv.2
	Y1003mut	Tepotinib; Capmatinib; Crizotinib		Lv.3
	Fusions	Crizotinib	All Solid Tumors	Lv.4
*MTOR*	E2014K, E2419K	Everolimus	Bladder Cancer	Lv.3
	Q2223K	Everolimus	Renal Cell Carcinoma	
	L2209V, L2427Q	Temsirolimus		
	Oncogenic Mutations	Everolimus; Temsirolimus	All Solid Tumors, Rhabdomyosarcoma ^c^	Lv.4
*NF1*	Oncogenic Mutations	Selumetinib	Neurofibroma ^b^	Lv.1
		Trametinib; Cobimetinib	All Solid Tumors	Lv.4
*NRG1*	Fusions	Zenocutuzumab	All Solid Tumors	Lv.3
*NTRK1/2/3*	Fusions	Entrectinib; Larotrectinib	All Solid Tumors ^b^	Lv.1
*PALB2*	Oncogenic Mutations	Olaparib	Prostate Cancer	Lv.1
*PDGFB*	COL1A1-PDGFB Fusion	Imatinib	Dermatofibrosarcoma Protuberans	Lv.1
*PDGFRA*	Exon 18 in-frame deletions or insertions, Exon 18 missense mutations	Avapritinib	Gastrointestinal Stromal Tumor	Lv.1
	Oncogenic Mutations	Regorafenib	Gastrointestinal Stromal Tumor; Medullary thyroid cancer ^c^, Hepatocellular carcinoma^c^	Lv.2
		Imatinib; Ripretinib; Sunitinib	Gastrointestinal Stromal Tumor	
	D842V	Dasatinib		
	D842V	Imatinib	Gastrointestinal Stromal Tumor	Lv.R1
*PIK3CA*	C420R and 10 other alterations	Alpelisib + Fulvestrant	Breast Cancer	Lv.1
	Oncogenic Mutations (excluding C420R, E542K, E545A, E545D, E545G, E545K, Q546E, Q546R, H1047L, H1047R and H1047Y)	Alpelisib + Fulvestrant		Lv.2
*PTCH1*	Truncating Mutations	Sonidegib; Vismodegib	Medulloblastoma	Lv.3
*PTEN*	Oncogenic Mutations	GSK2636771; AZD8186	All Solid Tumors	Lv.4
*RAD51B*, *RAD51C*, *RAD51D*, *RAD54L*	Oncogenic Mutations	Olaparib	Prostate Cancer	Lv.1
*RET*	Fusions or Oncogenic Mutations	Pralsetinib; Selpercatinib	Non-Small Cell Lung Cancer, Thyroid Cancer, Medullary Thyroid Cancer ^b^	Lv.1
	Fusions	Cabozantinib	Non-Small Cell Lung Cancer; Sarcoma ^c^	Lv.2
		Vandetanib	Non-Small Cell Lung Cancer	Lv.3
*ROS1*	Fusions	Crizotinib	Non-Small Cell Lung Cancer	Lv.1
		Entrectinib	Biomarker (+), solid and brain ^b^	
*SMARCB1*	Deletion	Tazemetostat	Epithelioid Sarcoma	Lv.1
*STK11*	Oncogenic Mutations	Bemcentinib + Pembrolizumab	Non-Small Cell Lung Cancer	Lv.4
*TSC1/2*	Oncogenic Mutations	Everolimus	Encapsulated Glioma; Subependymal giant cell astrocytoma ^b^	Lv.1

^a^ FDA-approved level 1 = FDA-recognized biomarker predictive of response to an FDA-approved drug in this indication; level 2 = Standard care biomarker recommended by the NCCN or other professional guidelines predictive of response to an FDA-approved drug in this indication; level 3 = Standard care or investigational biomarker predictive of response to an FDA-approved or investigational drug in another indication; level 4 = Compelling biological evidence supports the biomarkers as being predictive of response to a drug; level R1 = Standard care biomarker predictive of resistance to an FDA-approved drug in this indication; level R2 = Compelling clinical evidence supports the biomarker as being predictive of resistance to a drug. ^b^ FDA-approved for pediatrics used [61]. ^c^ Clinical trial in pediatrics.

**Table 3 cancers-15-01418-t003:** Somatic and germline mutated genes of selected pediatric tumors.

Tumor	Significantly Mutated Genes (^#^ Prevalence)
Medulloblastoma	DDX3X (5.8%), KMT2D (5.8%), CTNNB1 (5.5%), PTCH1 (5.1%), TP53 (4.0%), SMARCA4 (3.6%), KDM6A (3.1%), SUFU (1.3%), SMO (1.5%), KMT2C (1.4%), CREBBP (1.3%), APC ^†^ (0.6%), IDH1 (0.4%)
High grade glioma	*TP53*^†‡^*(28.5%)*, *ATRX (11.3%)*, *PIK3CA (5.6%)*, *PDGFRA*^‡^*(5.1%)*, *BCOR (3.0%)*, *PPM1D*^‡^*(3.9%)*, *CREBBP*^‡^*(1.8%)*, *NF1*^†^*(0.8%)*, *EGFR*^‡^*(0.6%)*
Ependymoma	*RELA*^‡^*(25.0%)*, *IGF2R*^†^*(20.0%)*
Low grade glioma	*FGFR1*^‡^*(33.3%)*, *BRAF (8.7%)*, *NF1*^†^*(3.9%)*, *KIAA1549 (1.9%)*
Neuroblastoma	*MYCN (36.2%)*, *MYCNOS (33.0%)*, *ATRX (22.2%)*, *DDX1 (22.3%)*, *ALK (1.4%)*, *RYR1 (0.5%)*, *PTPN11 (0.7%)*
Wilms tumor	*MYCN (12.4%)*, *MYCNOS (12.4%)*, *TP53 (3.2%)*, *DROSHA*^‡^*(1.8%)*, *WT1 (1.6%)*, *CTNNB1 (1.5%)*, *DGCR8 (1.1%)*
Osteosarcoma	*TP53*^†^*(30.0%)*, *RB1*^†^*(15.4%)*, *ATRX (9.7%)*
Ewing’s sarcoma	*EWSR1 (29.6%)*, *FLI1 (25.9%)*, *ERG (4.7%)*, *STAG2 (2.4%)*
Retinoblastoma	*RB1*^†^*(51.6%)*, *BCOR (3.2%)*
Rhabdomyosarcoma	*PAX3*^‡^*(28.6%)*, *FOXO1*^‡^*(25.9%)*, *PAX7*^‡^*(16.7%)*, *TP53*^†‡^*(12.3%)*, *FGFR4*^‡^*(7.7%)*, *NRAS*^‡^*(4.6%)*

^#^ Prevalence of mutated genes in the selected pediatric tumor. Data from cBioPortal for cancer genomics (www.cbioportal.org; accessed on 30 April 2022). ^†^ Germline, ^‡^ Relapse. Data from St. Jude Cloud public data repository (www.stjude.cloud; accessed on 18 September 2022).

**Table 4 cancers-15-01418-t004:** Significant genomic alterations of actionable genetic mutations in pediatric solid tumors.

Signaling Pathway	Gene	Alterations	Effected Domain	Pediatric CANCER Types	Potentially Targeted Therapy (Level of Evidence)	Additional References for Targeted Therapy
Tyrosine Kinase	*ALK*	Fusion		NBL	Crizotinib, Ceritinib, Alectinib, Lorlatinib	cBioPortal
		F1174L ^‡^	CAD exon23	NBL	Crizotinib (B)	[74,75]
		F1245V	CAD exon24	NBL		
		R1275Q/L ^†‡^	CAD exon25	NBL		
	*NTRK1*	TPM3::NTRK1		HGG	Larotrectinib (A)	[18,76,77]
	*NTRK2*	Fusion		HGG, LGG	Larotrectinib (A)	[77,78,79]
	*NTRK3*	ETV6::NTRK3		HGG, LGG	Larotrectinib (A)	[76,77]
	*PDGFRA*	Y288C	Exon6	HGG	Imatinib, sunitinib, regorafenib and ripretinib	cBioPortal
		E311_E7splice	Exon7	HGG		
		N659K ^‡^	PKD exon14	HGG	Imatinib, sunitinib, regorafenib and ripretinib	cBioPortal
		D842Y	PKD exon18	HGG	Avapritinib, Imatinib, Sunitinib	cBioPortal
	*ROS1*	Fusion		OS, HGG	Crizotinib, Entrectinib	cBioPortal
MAPK signaling	*NF1*	Fusion		OS, NBL, MB, HGG	Trametinib, Cobimetinib	cBioPortal
		Mutation		LGG, NBL	Selumetinib (B)	[80,81,82]
	*BRAF*	KIAA1549::BRAF		LGG, PA	Selumetinib (B), Sorafenib (C)	[81,83,84]
		V600E		LGG, HGG, PA, NBL	Selumetinib (B), Vemurafenib (B), Dabrafenib (B)	[81,85,86]
	*KRAS*	G12D	GTPase exon2	LGG, NBL	Trametinib, Cobimetinib, Binimetinib	cBioPortal
	*NRAS*	G12S	GTPase exon2	HGG	Binimetinib, Binimetinib + Ribociclib	cBioPortal
		Q61K ^‡^/R	GTPase exon3	RHB, NBL		
	*PTPN11*	E69K	Exon3	NBL, PA		
		A72T/D	Exon3	NBL		
		E76A	Exon3	NBL, PA		
Notch signaling	*NOTCH2*	Fusion		OS, NBL		
		R5_P6fs	Exon1	OS, NBL, RHB		
		P6fs	Exon1	NBL, MB, PA, WLM		
Sonic hedgehog signaling	*PTCH1*	Mutation		MB	Sonidegib (B)	[87]
		A300fs	Exon6	MB	Sonidegib, Vismodegib	cBioPortal
		Y804fs	Exon15	MB	Sonidegib, Vismodegib	cBioPortal
	*SMO*	L412F		MB	Vismodegib ^#^ (C)	[88]
		W535L		MB	Vismodegib ^#^	cBioPortal
Wnt signaling	*CTNNB1*	D32	Exon3	MB		
		S33	Exon3	MB		
		G34	Exon3	MB, RHB, ACT, HB		
		S37	Exon3	MB		
		T41A/N	Exon3	WLM, MB, RHB		
		N387K ^‡^	Exon8	WLM		
PI3K signaling	*PTEN*	Fusion		OS	GSK2636771, AZD8186	cBioPortal
		R130	CAD exon5	HGG		
		R233 *	Exon7	HGG		
	*PIK3CA*	R88Q	SBD exon2	HGG	Alpelisib + Fulvestrant	cBioPortal
		N345K ^‡^	Exon5	MB, RHB, EPD		
		E545K	Exon10	HGG		
		Q546K	Exon10	HGG, MB		
		E888 *	CAD exon18	NBL		
		H1047R/L	CAD exon21	HGG, MB, RHB, NBL		
	*FGFR1*	Fusion		LGG	Erdafitinib, Infigratinib	cBioPortal
		Internal tandem duplication	CAD	LGG		
		N546K	CAD exon12	LGG, NBL, PA, WLM, HGG	Pemigatinib (C)	[89]
		K656E	CAD exon14	PA, HGG, WLM	Erdafitinib, Infigratinib	cBioPortal
	*FGFR4*	V550L ^‡^	CAD exon13	RHB		
	*EGFR*	A289V	Exon7	HGG	Lapatinib	cBioPortal
TGFB signaling	*ACVR1*	R206H	CAD exon6	HGG		
		R258G	CAD exon7	HGG		
		G328E/V	CAD exon8	HGG		
		G356_E9splice	CAD exon9	HGG		
Cell cycle and DNA repair	*RB1*	Fusion		OS		
		W78 *	Exon2	OS		
		R320 *	Exon10	RB, HGG		
		R445 *^†^	Exon14	RB		
		R552 *	Exon17	RB, OS, HGG		
		R579 *	Exon18	RB		
	*TP53*	Mutation		HGG, WLM, OS, MB	Vismodegib (C)	[90]
		T125T/R ^†^	DBD exon4	HGG, WLM, ACT		
		R175H ^†‡^	DBD exon5	HGG, WLM, MB, RHB, ACT		
		C176F	DBD exon5	RHB, EWS, NBL		
		R213 *^†^	DBD exon6	HGG, MB		
		G245S	DBD exon7	HGG, MB		
		R248Q/W ^†^	DBD exon7	MB, HGG, OS, WLM		
		R273C ^†^/H	DBD exon4	HGG, EWS, ACT, MB, OS		
		R282W ^†^	DBD exon8	OS, HGG, MB		
		R337H ^†^	Exon10	ACT		
		R342 */P	Exon10	HGG, WLM		
	*CDK1*	V124G	CAD exon5	MB		
	*PPM1D*	W427 *	Exon6	HGG		
		S516 *	Exon6	HGG, NBL		
		E525 *	Exon6	HGG, MB		
Transcriptional regulation	*EWSR1*	FLI1::EWSR1		EWS	TK216	cBioPortal
		ERG::EWSR1		EWS		
	*BCOR*	R1164*	Exon7	HGG		
		H1481fs	Exon11	HGG		
	*SIX1*	Q177R	DBD exon1	WLM		
	*MYCN*	Fusion		NBL		
		P44L	Exon2	WLM, NBL, MB		
	*PAX7*	FOXO1::PAX7		RHB		
	*PAX3*	FOXO1::PAX3		RHB		
RNA processing	*DROSHA*	E1147K	Ribonuclease exon29	WLM		
		D1151	Ribonuclease exon29	WLM, NBL		
	*DGCR8*	E518K	RBM exon7	WLM		
	*DDX1*	DDX1::DDX1		NBL		
		MYCN::DDX1		NBL		
	*DDX3X*	R351W	HD exon11	MB		
		M380I	HD exon11	MB		
		R534	HD exon14	MB		
Epigenetics	*ATRX*	ATRX::ATRX		NBL		
		N294fs	Exon9	OS		
	*ASXL1*	R643fs	Exon13	WLM		
		R693 *	Exon13	HGG, EPD		
	*H3-3A* *(H3F3A)*	K28M	Exon2	HGG, LGG		
		G35R	Exon2	HGG		
	*KMT2C*	T1636P	Exon33	MB		
		E2798fs	Exon38	MB		
		I4084L	Exon48	MB		
	*SMARCA4*	T910M	HD exon19	MB		
	*H3C2* *(HIST1H3B)*	K28M ^‡^	Exon1	HGG		
	*KDM6A*	S54_E2splice	Exon2	MB		
		R1351 *	Exon28	MB		
	*IDH1*	R132C/H	Exon4	MB, HGG, LGG	Bevacizumab and Sunitinib (B)	[91]
		R222C/H	Exon6	HGG, EWS		
	*RELA*	Fusion		EPD, HGG		
	*STAG2*	R216 *	STAG domain exon8	EWS		
		R259 *	STAG domain exon9	MB, HGG		
		E1209Q	Exon33	OS		
	*FLI1*	EWSR1::FLI1		EWS		
	*ERG*	EWSR1::ERG		EWS		

^†^ Germline, ^‡^ Relapse, ^#^ Reduce treatment activity, * Termination codon. Abbreviations: ACT, adrenocortical carcinoma; CAD, Catalytic domain; ECD, extracellular domain; DBD, DNA binding domain; EPD, ependymoma; EWS, Ewing sarcoma; HB, hepatoblastoma; HD, Helicase domain; HGG, high grade glioma; LGG, low grade glioma; MB, medulloblastoma; NBL, neuroblastoma; OS, osteosarcoma; PA, pilocytic astrocytoma; PKD, Protein kinase domain; RB, retinoblastoma; RBM, RNA binding motif; RHB, rhabdosarcoma; SBD, Substrate binding domain; WLM, Wilms’ tumor; Level of evidence: A, validated association; B, clinical evidence; C, case study; D, preclinical evidence; E, inferential association.

**Table 5 cancers-15-01418-t005:** Precision study designs for pediatric cancer: Single-arm design.

Gene Involved in Trial Design	NCT (Recruitment Status)	Phase	Specification	Intervention(s)	Cancer Type(s)	Eligibility	Enrollment (Number)
*ALK*	NCT01742286 ^(D)^	I	ALK alterations	Ceritinib	ALK-activated Tumors	1–17 years	83
	NCT02465528 ^(C)^	II	ALK alterations	Ceritinib	Tumors With Aberrations in ALK, Glioblastoma	≥18 years	22
	NCT02780128 ^(A)^	I	ALK mutation	Ceritinib + Ribociclib	Neuroblastoma	1–21 years	131
	NCT03107988 ^(A)^	I	ALK alterations	Lorlatinib + Chemotherapy	Neuroblastoma	≥1 year	65
	NCT03194893 ^(B)^	III	ALK alterations	Alectinib or Crizotinib	Neoplasms	all	200
	NCT04774718 ^(A)^	I, II	ALK fusion	Alectinib	ALK Fusion-positive Solid or CNS Tumors	≤17 years	42
	NCT05384626 ^(A)^	I, II	ALK alterations	NVL-655	Solid Tumor, NSCLC	≥12 years	214
*BRAF*	NCT01089101 ^(B)^	I, II	BRAF V600E mutation or BRAF-KIAA1549 fusion	Selumetinib	Low Grade Glioma, Recurrent Childhood Pilocytic Astrocytoma, Recurrent Neurofibromatosis Type 1	3–21 years	220
	NCT01596140 ^(D)^	I	BRAF mutation	Vemurafenib + Everolimus or Temsirolimus	Advanced Cancer, Solid Tumor	all	27
	NCT01636622 ^(D)^	I	BRAF mutation	Vemurafenib + Chemotherapy	Advanced Cancers	≥12 years	21
	NCT01677741 ^(D)^	I, II	BRAF V600 mutation	Dabrafenib	Neoplasms, Brain	1–17 years	85
	NCT02124772 ^(D)^	I, II	BRAF V600 mutation	Dabrafenib + Trametinib	Solid Tumors, neuroblastoma, low grade glioma, neurofibromatosis Type 1	1 month to 17 years	139
	NCT02684058 ^(B)^	II	BRAF V600 mutation	Dabrafenib + Trametinib + Radiation	Solid Tumors, CNS Tumors, high grade glioma, low grade glioma	1–17 years	149
	NCT03919071 ^(A)^	II	BRAF V600 mutation	Dabrafenib + Trametinib + Radiation	Anaplastic Astrocytoma, Glioblastoma, Malignant Glioma	1–21 years	58
	NCT04576117 ^(A)^	III	BRAF rearrangement	Selumetinib + Chemotherapy	Low Grade Astrocytoma, Glioma	2–25 years	18
*EGFR*	NCT00198159 ^(C)^	II	EGFR expression	Gefitinib	Refractory Germ Cell Tumors Expressing EGRF	≥15 years	21
	NCT00418327 ^(D)^	I	EGFR mutation	Erlotinib + Radiation	Malignant Brain Tumor, Glioma	1–21 years	48
	NCT01182350 ^(C)^	II	EGFR overexpression	Erlotinib + Bevacizumab + Temozolomide + Radiation	Diffuse Intrinsic Pontine Glioma	3–18 years	53
	NCT01962896 ^(C)^	II	EGFR/mTOR pathway activation	Erlotinib + Sirolimus	Relapsed/Recurrent Germ Cell Tumors	1–50 years	4
*EWSR1*	NCT03709680 ^(A)^	II	EWSR1-ETS or FUS-ETS rearrangement	Palbociclib + Chemotherapy	Ewing Sarcoma, Rhabdomyosarcoma, Neuroblastoma, Medulloblastoma, Diffuse Intrinsic Pontine Glioma	2–20 years	184
	NCT04129151 ^(B)^	II	EWSR1 or FUS translocation	Palbociclib + Ganitumab	Ewing Sarcoma	12–50 years	18
*FGFR*	NCT04083976 ^(A)^	II	FGFR alteration	Erdafitinib	Advanced Solid Tumor	≥6 years	336
	NCT05180825 ^(A)^	II	FGFR1 and MYB/MYBL1 alterations, 7q34 duplication	Trametinib or Vinblastine	Grade 1 Glioma, Mixed Glio-neuronal Tumors, Pleomorphic Xanthoastrocytoma	1 month to 25 years	134
*H3*	NCT02525692 ^(B)^	II	H3 K27M mutation	ONC201	Glioblastoma, Glioma	≥16 years	89
	NCT03416530 ^(A)^	I	H3 K27M mutation	ONC201	Diffuse Intrinsic Pontine Glioma, Glioma, Malignant	2–18 years	130
	NCT05009992 ^(A)^	II	H3 K27M mutation	ONC201 + Paxalisib or Radiation	Diffuse Intrinsic Pontine Glioma, Diffuse Midline Glioma, H3 K27M-Mutant	2–39 years	216
*IDH*	NCT03749187 ^(A)^	I	IDH1/2 mutation	PARP Inhibitor BGB-290 + Chemotherapy	Glioblastoma, Glioma	13–39 years	78
*MYCN*	NCT02559778 ^(A)^	II	MYCN amplification	Ceritinib, Dasatinib, Sorafenib or Vorinostat + Chemotherapy	Neuroblastoma	≤22 years	500
	NCT03126916 ^(A)^	III	MYCN amplification	Lorlatinib + Standard therapy	Ganglioneuroblastoma, Neuroblastoma	1–30 years	658
*NF*	NCT01158651 ^(D)^	II	NF1 mutation	Everolimus	Glioma	1–21 years	23
	NCT03095248 ^(A)^	II	NF2 mutation	Selumetinib	Neurofibromatosis 2, Vestibular Schwannoma, Meningioma, Ependymoma, Glioma	3–45 years	34
	NCT03326388 ^(A)^	I, II	NF1 positive	Selumetinib	Neurofibromatosis Type 1, Plexiform Neurofibroma, Optic Nerve Glioma	3–18 years	30
	NCT03871257 ^(A)^	III	NF1 positive	Selumetinib + Chemotherapy	Low Grade Glioma, Neurofibromatosis Type 1, Visual Pathway Glioma	2–21 years	290
*NTRK*	NCT02637687 ^(A)^	I, II	NTRK-fusion	Larotrectinib	Solid Tumors Harboring NTRK Fusion	≤21 years	155
	NCT03834961 ^(A)^	II	NTRK-fusion	Larotrectinib	Solid Tumor, CNS Tumor	≤30 years	70
	NCT04879121 ^(A)^	II	NTRK amplification	Larotrectinib	Solid Neoplasm	≥16 years	13
*PDGFR*	NCT00417807 ^(D)^	I, II	PDGFR expression	Imatinib	Refractory Desmoplastic Small Round Cell Tumors	≥16 years	9
	NCT03352427 ^(C)^	II	PDGFR alteration	Dasatinib + Everolimus	Glioma, High Grade Glioma, Pontine Tumors	1–50 years	3
*Rb1*	NCT02255461 ^(C)^	I	Rb1 positive	Palbociclib	CNS Tumors, Solid Tumors	4–21 years	35
	NCT03355794 ^(B)^	I	Rb1 positive	Everolimus + Ribociclib	Diffuse Intrinsic Pontine Glioma, Malignant Glioma of Brain, High Grade Glioma, Glioblastoma, Anaplastic Astrocytoma	1–30 years	24
	NCT03387020 ^(D)^	I	Rb1 positive	Everolimus + Ribociclib	CNS Tumors	1–21 years	22
*ALK* *c-MET* *ROS*	NCT00939770 ^(D)^	I, II	ALK or MET alterations	Crizotinib	Recurrent Neuroblastoma	1–21 years	122
	NCT01524926 ^(B)^	II	ALK or MET pathway activation	Crizotinib	Lymphoma, Sarcoma, Rhabdomyosarcoma	≥1 year	582
	NCT02034981 ^(B)^	II	ALK, MET or ROS1 alterations	Crizotinib	Solid Tumors	≥1 year	246
	NCT02650401 ^(A)^	I, II	ALK, ROS1, or NTRK1-3 Rearrangements	Entrectinib	Solid Tumors, CNS Tumors, Neuroblastoma	≤18 years	68
	NCT03093116 ^(A)^	I, II	ALK, ROS1, or NTRK1-3 Rearrangements	Repotrectinib	Solid tumor, CNS tumor	≥12 years	500
*RAS* *RAF* *MEK* *ERK* *NF1*	NCT02285439 ^(B)^	I, II	BRAF truncated fusion or NF1 mutation	MEK162	Low-Grade Gliomas, Brain, Soft Tissue Neoplasms	1–18 years	105
	NCT02639546 ^(D)^	I, II	RAS/RAF/MEK/ERK pathway activation	Cobimetinib	Solid Tumors	6 months to 30 years	56
	NCT03363217 ^(A)^	II	BRAF-KIAA1549 fusion, NF1 mutation, MAPK/ERK pathway activation	Trametinib	Low-grade Glioma, Plexiform Neurofibroma, Central Nervous System Glioma	1 month to 25 years	150
	NCT04201457 ^(A)^	I, II	BRAF V600 mutation or truncated fusion, NF1 mutation	Dabrafenib + Trametinib + hydroxychloroquine	Low Grade Glioma, High Grade Glioma	1–30 years	75
	NCT04216953 ^(A)^	I, II	MAPK pathway status and Tumor Mutational Burden	Cobimetinib + Atezolizumab	Sarcoma, Soft Tissue	≥6 months	120
*SHH* *WNT*	NCT00822458 ^(D)^	I	SHH or WNT signaling activation	Vismodegib	Recurrent Childhood Medulloblastoma	3–21 years	34
	NCT01239316 ^(D)^	II	SHH signaling activation	Vismodegib	Recurrent Childhood Medulloblastoma	3–21 years	12
	NCT01878617 ^(A)^	II	SHH or WNT signaling activation	Vismodegib + chemotherapy	Medulloblastoma	3–39 years	660
*Others*	NCT01396408 ^(B)^	II	Mutations in sunitinib targets such as VEGFR, PDGFR, KIT, RET or mutations in mTOR pathway such as PTEN, TS1/2, LKB1, NF1/2	Sunitinib or temsirolimus	Advanced Rare Tumors	≥16 years	137
	NCT03654716 ^(A)^	I	MDM2, MDMX, PPM1D or TET2 amplification	ALRN-6924	Solid Tumor, CNS Tumor	1–21 years	69

Recruitment status: ^(A)^ Recruiting, ^(B)^ Active, not recruiting, ^(C)^ Terminated, ^(D)^ Completed.

**Table 6 cancers-15-01418-t006:** Precision study designs for pediatric cancer: Multiple-arm design.

Gene Involved in Trial Design	NCT (Recruitment Status)	Phase	Specification	Intervention(s)	Cancer Type(s)	Eligibility	Enrollment (Number)
Testing the Use of Food and Drug Administration (FDA)-Approved Drugs (TAPUR)	NCT02693535 ^(A)^	II	ALK, ROS1, MET	Crizotinib	Advanced Solid Tumors	≥12 years	3581
			CDKN2A, CDK4, CDK6	Palbociclib or Abemaciclib			
			CSF1R, PDGFR, VEGFR	Sunitinib			
			mTOR, TSC	Temsirolimus			
			BRAF V600E/D/K/R	Vemurafenib and Cobimetinib			
			RET, VEGFR1/2/3, KIT, PDGFRβ, RAF-1, BRAF	Regorafenib			
			BRCA1/2, ATM	Olaparib			
			NRG1	Afatinib			
			BRCA1/2, PALB2	Talazoparib			
			ROS1 fusion	Entrectinib			
			NTRK amplification	Larotrectinib			
NCI-COG Pediatric MATCH Screening	NCT03155620 ^(A)^	II	NTRK1, NTRK2, or NTRK3 gene fusion	Larotrectinib	Refractory or Recurrent Advanced Solid Tumors	1–21 years	2316
			FGFR1, FGFR2, FGFR3, or FGFR4 gene mutation	Erdafitinib			
			EZH2, SMARCB1, or SMARCA4 gene mutation	Tazemetostat			
			TSC1, TSC2, or PI3K/mTOR gene mutation	Samotolisib			
			activating MAPK pathway gene mutation	Selumetinib			
			ALK or ROS1 gene alteration	Ensartinib			
			BRAF V600 gene mutation	Vemurafenib			
			ATM, BRCA1, BRCA2, RAD51C, RAD51D mutations	Olaparib			
			Rb positive, alterations in cell cycle genes	Palbociclib			
			MAPK pathway mutations	Ulixertinib			
			HRAS gene alterations	Tipifarnib			
			RET activating mutations	Selpercatinib			
TAPISTRY Platform Study	NCT04589845 ^(A)^	II	ROS1 fusion	Entrectinib	Solid Tumor	all	770
			NTRK1/2/3 fusion	Entrectinib			
			ALK fusion	Alectinib			
			AKT1/2/3 mutation	Ipatasertib			
			PIK3CA multiple mutation	Inavolisib			
			BRAF mutation or fusion-positive	Belvarafenib			
			RET fusion-positive	Pralsetinib			

Recruitment status: ^(A)^ Recruiting.

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
