# Peer review of "Genomics-Driven Precision Medicine in Pediatric Solid Tumors"

_cancers, 2023, doi:10.3390/cancers15051418_

Round 1

Reviewer 1 Report

The review from Suthapot and colleagues is nicely written and flows really well. They provide an updated overview of the status of precision medicine in pediatric oncology. They have put together informative tables with the most recent data regarding treatments.

I would highly recommend this review for publication.

Reviewer 2 Report

The authors provided a comprehensive overview of genomic-driven precision medicine.  There are few points I like to point out, and authors can decide if they want to address/ include them 

1. Many pediatric cancers are rare, and finding the right patient population for the drugs is challenging. The following paper was published in nature reviews Fletcher, J.I. et al. Too many targets, not enough patients: rethinking neuroblastoma clinical trials. Nat Rev Cancer 2018, emphasizes this.

2. Pediatric genome is less complicated than the adult genome, so genomic targeting is easier compared to adults 

3. Drug tolerability in Pediatric patients is less than in adults, so drugs cannot achieve preferred efficacious levels compared to others 

4. The paper requires extensive English editing, ex: B was not italicized and hypen was missing between catenin in table 1

Reviewer 3 Report

In this manuscript, Suthapot et al reviewed the current knowledge on genomic mutations and alterations in different paediatric tumours, which are less well investigated than adult tumours. The authors summarized the different genes found to be mutated in paediatric cancers and whether treatments already in used in adult tumours could be used in paediatric tumours. The final part of the review described the different clinical trials currently underway for paediatric cancers.

This is an exhaustive and well-organised review on the field of genomic precision medicine in paediatric tumours, which will be of interest to numerous researchers.

I have only a few minor comments:

1.       Table 2: The c is not described in the legend.

2.       Table 2: Sometimes it is written Lv. R1 and Lv. R2, is it the same as Lv.1 and Lv. 2 or does the R means something?

3.       Figure 1: Some of the genes are in colour boxes, does the colour indicate something?

Reviewer 4 Report

The review entitled 'Genomics-driven Precision Medicine in Paediatric Oncology’

 The review article is very intriguing, and details the latest trials and advancements in the field. The introduction section is very engaging, the authors nicely described the molecular mechanisms and their role in various cancers.

The authors could improve the review article.

1.      The article focussed on inhibitors, there are other advanced therapies like CAR-T and Immunotherapy with emerging modalities like neo antigens etc. The authors need to shed light on these topics.

2.      The authors briefly described about mRNA therapy but they need to elaborate as there are clinical trails ongoing on personalized cancer vaccine, and if that could be applicable to paediatric cancers.

3.      The tables described are broadly describing targeting genes related to various cancers but their prevalence in paediatric population is not highlighted. Need more details in that context.

4.      Abbreviations need to be described like NTRK,CNS, RTK, MAP.  

5.      Line 383, the authors described as ‘lunch’ instead of launch.

6.      The authors need a section on emerging targets in immunotherapy.

7.      With the emergence of gene editing technologies like TALENS, CRISPR-CAS9, the authors could discuss about it.

8.      The discussion is too short, can the authors discuss more broadly on the emerging targets and their implications in cancer.

9.      The abbreviations can be included in the end as a separate section. 

10.  References are missing on page 13 for the targets described in sections like RNA processing, transcriptional regulation etc.

Reviewer 5 Report

The review article entitled “Genomics-driven Precision Medicine in Pediatric Oncology,” is a very comprehensive compilation of pediatric-based molecular-guided therapies. While the article is very interesting there are several revisions that may be needed to help improve clarity.

1.       How many percentage of the germline mutations of a particular gene result in in the solid tumors present in Table 1? For instance, what percentage of RB1 germline mutations results in osteosarcoma?

2.       It is not stated if the significantly mutated genes listed in Table 3 also impact protein? Are these mutations impacting function?

3.       It would be good to include some data on the outcomes of a few of the clinical trials presented in the review.

4.       For patients placed on genomics driven therapy what is the success rate? Does it increase progression free survival, etc?

5.       Many of the tables run over to the next page which is understandable, however, it would be great if the labels of each column on each table can be repeated every time the table moves on to the next page so it makes it easy for the readers.

6.       Please define what he abbreviations Lv.1, Lv.2, Lv.3, or Lv.4 mean

7.       In table 2 where amplifications are listed, does the also result in increased protein expression?

8.       In table 2 where insertions and or deletions are listed please elaborate on how they may impact protein function.

9.       Figure 1 should also include pediatric cancers that are genomically complex to the point that a single oncogenic drivers do not exist but rather multiple oncogenic drivers play a role such as osteosarcoma. Also include Wilms, and other forms of rhabdomyosarcoma.

10.   Table 3 rhabdomyosarcoma is misspelled “rhabdosarcoma”

11.   In table 4 for each gene please elaborate on whether the effected domains impact function or are the genomic alterations not impacting function. Will they result in gain-of-function or loss-of-function impact?

12.   Table 4 also includes gliomas, glioblastoma – how common are these in pediatric population.

13.   The title of the review is “Genomics -drive Precision Medicine in Pediatric Oncology,” but it seems the review mostly discuss pediatric solid tumors so may be the title should incorporate solid tumors into it. Otherwise, genomics guided therapy in pediatric liquid tumors should be discussed in depth here as well.

14.   It may be also good to discuss toxicities of some of the drugs used in adult cancers vs pediatric cancers.

Round 2

Reviewer 5 Report

None